

# Automated Analysis and Quality Assurance of Ice-Nucleating Particle Data: The PINE INP Analysis Software PIA

Nicole Büttner[1], Romy Fösig[1], Alexander Böhmländer[1], Larissa Lacher[1], Franziska Vogel[1a], Mark Tarn[2], Pia Bogert[1], Jens Nadolny[1], Benjamin Murray[2], and Ottmar Möhler[1]

[1]Institute of Meteorology and Climate Research, Karlsruhe Institute of Technology, Karlsruhe, Germany
[a]Now at: Institute of Atmospheric Sciences and Climate (ISAC), National Research Council (CNR), Bologna, Italy
[2]School of Earth and Environment, University of Leeds, Leeds, UK

**Correspondence:** Nicole Büttner (nicole.buettner@kit.edu)

**Abstract.** The presence of ice-nucleating particles (INPs) in the atmosphere plays a crucial role in shaping cloud radiative properties, influencing their lifespan, and affecting precipitation and storm dynamics. To enable continuous and high-resolution monitoring of INP concentrations, the Portable Ice Nucleation Experiment (PINE) was developed. Complementing this, the PINE INP Analysis (PIA) software was created to ensure a standardised and reproducible data processing workflow. This work presents the setup of software version 3.0.0 and the structure of the processed data. The two main components of the software - the automated quality control of the data and the algorithm to distinguish between aerosols and droplets versus ice crystals based on their optical size - are described in detail. The second part of this study provides recommendations for quality assurance of PINE measurements. It outlines procedures for conducting background checks to detect potential contamination within the chamber, evaluates the consistency between adjacent temperature sensors, and discusses how large aerosol particles can impact measurement uncertainty.

## 1 Introduction

Ice-nucleating particles (INPs; Vali et al., 2015) catalyse the first formation of ice crystals in clouds and can thereby influence their radiative properties (e.g., Murray et al., 2021). An increase in the INP concentration can lead to faster cloud glaciation and associated dissipation, which in turn reduces the cloud lifetime (e.g., Lohmann and Feichter, 2005). Moreover, precipitation occurrence and rates, especially over land and in extratropical regions, are strongly influenced by ice processes (Mülmenstädt et al., 2015; Field and Heymsfield, 2015; Heymsfield et al., 2020). In addition to other cloud microphysical processes, INPs have been shown to impact the formation of precipitation (e.g., Muhlbauer and Lohmann, 2009; Diehl and Grützun, 2018; Fan et al., 2017; Lin et al., 2022), and to impact storm development and dynamics (e.g., Yang et al., 2024; Chen et al., 2019; Hazra et al., 2022).

The scarcity of INPs, which are a very minor subset of atmospheric aerosol particles, challenges measurement techniques (Cziczo et al., 2017; DeMott et al., 2011). Ambient INP concentration measurements relevant for mixed-phase cloud conditions were started in the 1950s (e.g., Aufm Kampe and Weickmann, 1951; Bigg, 1957), but only in recent years long-term measurements using automated instruments became available. Continuous-flow diffusion chambers (CFDS) have been widely





used for short term INP field measurements over recent decades (e.g., Rogers, 1988; DeMott et al., 2010; Chou et al., 2011;
Lacher et al., 2017). CFDCs work by creating a region of supersaturation between two ice coated temperature controllable
walls. A temperature gradient results in a heat and water flux from the warm to cold wall that produces a region of defined
supersaturation. More recently, automated CFDC instruments were developed (Bi et al., 2019; Brunner and Kanji, 2021). Since
their initial development, CFDCs have been extensively utilised and continuously refined and validated.

More recently, a novel mobile and automated cloud expansion-type chamber was developed, the Portable Ice Nucleation Exper-
30 iment PINE (Möhler et al., 2021). Based on the design and working principle of the AIDA (Aerosol Interaction and Dynamics
in the Atmosphere; Möhler et al., 2003) simulation chamber, PINE establishes cloud-like conditions by rapidly expanding the
air inside the chamber, thus inducing the formation of supercooled water droplets and ice crystals at temperatures relevant to
mixed-phase clouds. PINE was used in several short- and long-term field campaigns to measure INPs that are active at mixed-
phase cloud temperatures (e.g., Lacher et al., 2024; Vogel et al., 2024; Canzi et al., 2025; Böhmländer et al., 2025). PINE is
35 commercially available, and due to its automated operation and minimal requirements for user input, it has a great potential
to become a key instrument for INP monitoring. For example, PINE is a reference instrument within the ACTRIS (Aerosol,
Clouds and Trace Gases Research Infrastructure; Laj et al., 2024) Topical Centre for Cloud In Situ Measurements (CIS).

An important part of making PINE a reliable monitoring instrument is good data management following the FAIR princi-
ples (Findability, Accessibility, Interoperability, and Reusability; Wilkinson et al., 2016). To establish a FAIR life cycle of the
40 data, the Python software PIA (PINE INP Analysis; Büttner and Fösig, 2024) was developed to ensure a standardised and
reproducible data analysis. This paper describes the individual steps of data processing for the different data levels. The most
important features of the software are the automated data quality control and the algorithm that automatically determines the
size threshold for distinguishing between aerosols and droplets versus ice crystals. The functionality of this algorithm, the
so-called ice threshold finder, is described in detail. Also included is a detailed overview of the quality control tests applied to
45 the data and the resulting handling of the flagged data. The conditions for the tests are based on instrument specifications, em-
pirical observations by the instrument operators, and statistical analysis. This study also investigates the characteristics of the
temperature sensors of the PINE instrument and provides recommendations for quality assurance. To show results created by
the software, this study includes a data set of a 4-month measurement period that was conducted close to Karlsruhe (Germany)
from 2020 to 2021.

## 2 PINE Setup and Description of Raw Data

### 2.1 PINE Setup

This section describes the setup and the working principles of the PINE instrument as manufactured by the company Bilfinger
Nuclear & Energy Transition GmbH (Würzburg, Germany). For a more detailed description see Möhler et al. (2021). The
abbreviations for the instrument sensors used by the control and the analysis software are shown in brackets. The PINE instru-
55 ment consists of an inlet system, a chamber with a cooling system, a particle detection system, and a control system. Upon
entering PINE via the aerosol inlet, the sampled air is guided through two parallel Nafion™ membrane diffusion dryers (Perma





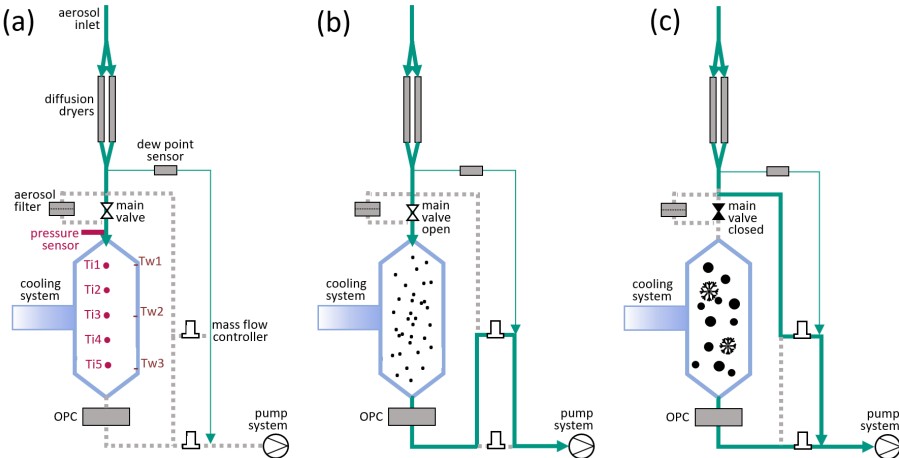

**Figure 1.** (a) Setup of PINE with the following main components: diffusion dryers, dew point sensor, main valve to the chamber, aerosol filter for background measurements, pressure sensor attached to the chamber, five gas temperature sensors (Ti1 – Ti5), three wall temperature sensors (Tw1 – Tw3), an optical particle counter (OPC) at the exit of the chamber, and a pump system controlling the different flows. The green lines indicate the active sample flow, while the grey dotted lines indicate no sample flow. The thin green line indicates the flow over the dew point sensor. Panel (b) shows the chamber during flush mode, the main valve is open and aerosol particles are flushed through the chamber. Panel (c) shows the expansion mode, the main valve is closed and cloud droplets and ice crystals can form. Figure adapted from Möhler et al. (2021).

Pure, MD-700-24S-1; Fig. 1(a)). By controlling the water vapour pressure gradient across the membrane, the humidity of the sampled air is reduced to the required measurement conditions. The humidity of the sampled air is measured with a dew point sensor (Vaisala, DRYCAP® DMT143) at ambient temperature in front of the chamber and is expressed as dew point tempera-
60 ture (DP). The DP of the sampled air needs to be higher than the temperature inside the chamber during expansion in order to allow the formation of cloud droplets, but at the same time, DP needs to be low enough to avoid frost formation that eventually can cause an ice background in the chamber. The sampled air enters the chamber via the main valve. The chamber has a volume of 10 L and is located within a vacuum chamber and cooled by a Stirling system. Five thermocouples (Thermoelement Typ K IKT 10/070 D2.5x15/2KK06/09 IEC 583-3 class 1, unknown manufacturer) located inside the cloud chamber (Ti1 – Ti5)
measure the gas temperature, and three pt-100 temperature sensors (Pt100 RS 1/5 class B, Sensorshop 24; Tw1 – Tw3) measure the temperature of the chamber wall. A pressure sensor attached to the inlet tubes upstream of the cloud chamber monitors the chamber pressure (Pch). An optical particle counter (OPC; fidas-pine, Palas GmbH) located directly below the cloud chamber determines the number and size of the particles within the sample flow.

PINE is operated in cycles of flushing the aerosol sample through the chamber (flush mode; Fig. 1(b)), expanding the air to
70 create cloud droplets and ice crystals (expansion mode; Fig. 1(c)), and refilling the chamber with filtered and dried air to balance the pressure to ambient conditions (refill mode). During the flush mode, the dried sample air is guided through the cooled cloud chamber at a constant mass flow rate in order to exchange the air inside the chamber. The photomultiplier (PM) voltage





of the OPC is tuned to the size range of large droplets and ice crystals. Therefore, it can not detect smaller aerosol particles. The expansion mode is started by closing the main valve upstream of the cloud chamber. The expansion flow (Fe) is set to a

constant volume flow rate, which is regulated by a mass flow controller (MFC; ELFLOW Select F-201CV 10 l, Bronkhorst Instruments GmbH; reference conditions are $T = 273.15\,\mathrm{K}$ and $p = 101\,325\,\mathrm{Pa}$) according to the current temperature and pressure conditions inside the chamber while the air is expanding. The INP concentration is therefore reported in number per standard litre ($\mathrm{stdL^{-1}}$). The cooling resulting from the decrease in pressure induces an increase in relative humidity. As soon as the saturation ratio in regard to water is exceeded, cloud droplets and ice crystals form in the presence of cloud condensation nuclei

and INPs, respectively. As aspherical ice crystals are optically larger than spherical cloud droplets for the OPC used in PINE, ice crystals can later be identified on the basis of their optical size (see Sect. 4.2). During expansion, the wall temperatures remain approximately constant and therefore cause an increasing heat flux from the wall to the gas. Due to this heat source, which leads to the evaporation of the cloud droplets inside the chamber, the cloud formation and ice crystal detection time is limited to approximately 40 seconds, depending on the expansion flow rate. The flow along the inlet system is kept constant

during the expansion by an additional bypass flow (see 1(c)). In the refill mode, ambient pressure conditions are established by refilling the chamber with filtered and dried air.

The number of formed ice crystals is related to the air volume analysed during the expansion. This allows the calculation of the ice crystal number concentration, which represents the INP concentration. The temperature assigned to the measured INP concentration is the minimum temperature (Ti5) recorded at the end of the expansion. Different variables are recorded during

the operations (Table 1), not only for determining the INP concentration, but also for data quality control measures (see Sect. 4.3).

The uncertainty budged of the PINE is discussed in Böhmländer et al. (2025). It can be concluded that the uncertainties are dominated by the uncertainty of the OPC, which is considered to be 10% of the INP concentration. For low INP concentrations statistical uncertainties given by

$$\frac{n_{\mathrm{INP}}}{\sqrt{N_{\mathrm{INP}}}},\tag{1}$$

where $n_{\mathrm{INP}}$ is the INP concentration and $N_{\mathrm{INP}}$ the INP count, become also relevant.

## 2.2   Raw Data

The PINE instrument is operated using an in-house developed LabVIEW control software that also manages the data acquisition. The control software creates a directory with the campaign name and stores multiple .txt files containing both measurement

data and metadata. If the campaign name changes, a new directory is generated automatically. Figure A1 shows the structure of the raw data. The PINE measurements are structured into runs and operations. One run is composed of one cycle of flush, expansion, and refill. One operation consists of multiple runs and is linked to certain measurement settings (e.g., temperature settings, filter for background measurements).

The files in the *operation setup* sub-directory contain the metadata from each operation. The relevant information for the anal-



**Table 1.** Overview of measurement variables.

| sensor | variable |
|---|---|
| gas temperature sensors | Ti1 - Ti5 |
| wall temperature sensors | Tw1 - Tw3 |
| main flow | Fm |
| expansion flow | Fe |
| dew point sensor | DP |
| pressure sensor | Pch |

ysis software is the ID of the OPC (*opc-id*) used in this setup and the name of the file used to map the OPC bin number to the particle diameter (*calib-file*).

The sub-directory *raw_Data* contains the files with the run-time information per operation. The names of these files start with *pfr* and will be referred to as *run files*. They also contain the pre-set flow rates and the end pressure during expansion. The sub-directory *opc-id* contains the raw data of the OPC per run. The file names end with *opc.txt* and will be referred to as *opc files*.

The sub-directory *housekeeping* contains files with the endings *valve.txt* and *instrument.txt* for each operation. The *valve.txt* files contain information about the opening and closing times of the valves and are not used for the analysis, but can be useful for troubleshooting. The *instrument.txt* files, referred to as *instrument files*, contain the data from the sensors inside the chamber at a 1 second resolution. The instrument files report the measurements of the following sensors: gas temperature 1 - 5 (Ti1 - Ti5), wall temperature 1 - 3 (Tw1 - Tw3), chamber pressure (Pch), main mass flow rate through the inlet (Fm), mass flow rate during expansion (Fe), and dew point temperature (DP).

The file starting with *pfo* will be referred to as *operation file*. It gives an overview of all operations in one campaign including the start and end times.

The data presented in this work were created with an older LabVIEW control software version that stored the run files in the *L0_Data* directory instead of *raw_Data*. The structure described here is that of the LabVIEW control software version 3.12.

An electronic logbook is needed for the analysis of the data. At the moment this logbook needs to be created manually as an Excel sheet. A template can be found within the software and should be stored under the campaign directory named *Logbook_<pine-id>_<campaign>.xlsx*. It needs to contain the operation type and the aerosol type (e.g. ambient air) for each operation. The operation type can be chosen from the following: background, constant temperature, temperature ramp, cirrus and test.





## 3 Data Set

The data set used in this study was created during the CORONA (Characterization Of Rural aerOsol and ice Nucleating pArticles) campaign that took place at Karlsruhe Institute of Technology (KIT) Campus North, close to Karlsruhe (Germany), from 2020 to 2022. The campaign was designed to monitor the influence of the pandemic and the corresponding measures on the atmospheric aerosol properties. The campus is located in a forest approximately 10 km north of Karlsruhe at 49° 5' 42.61'' N, 8° 25' 45.91'' E, 114 m above sea level. During this campaign, multiple aerosol instruments (e.g. Condensation Particle Counter (CPC), Scanning Mobility Particle Sizer (SMPS), and Aerodynamic Particle Sizer (APS)) were deployed and filter samples were collected for offline analysis with INSEKT (Ice Nucleation Spectrometer of the Karlsruhe Institute of Technology; described in Schneider et al. (2021)). All instruments were connected to the same inlet for sampling ambient air through a PM10 sampling head. Some results from the INSEKT measurements are described in Vogel (2022). Throughout the whole campaign period, different PINE instruments were connected to the inlet. The data used here were measured by the PINE version 04-02 (PINE-04-02) during the periods 16 November 2020 - 25 February 2021 and 18 March 2021 - 22 April 2021, providing roughly four months of data in total. The first period will be referred to as CORONA campaign and the second part will be referred to as CORONA_new campaign. During these time periods, PINE was mainly operated at a constant temperature between -19 °C and -20 °C. Occasionally, temperature ramps were performed in a temperature range between -16 °C and -27 °C to extend the measured INP data towards higher and lower temperatures. The mass flow rate during flush (Fm) was varying between 2 and 3 stdL min$^{-1}$ and the expansion volume flow rate was set to 3 L min$^{-1}$. The pressure inside the chamber was reduced by 150 - 200 mbar during the expansion.

## 4 PIA Software

### 4.1 Structure and Data Flow

The PINE INP Analysis (PIA) software is a tool written in Python, that allows for standardised analysis of PINE measurements. This work describes the structure and functionality of PIA v3.0.0. It consists of five sections, represented as classes (see Fig. 2) and are based on a data structure with processing levels 0 - 2. The first class deals with Level 0 data, stored in *level0_data.py*. The second and third classes are processing Level 1 and Level 2 data, respectively, and are stored in the *level_data.py* file. After that, the data undergo automated quality control tests, and finally, overview and analysis plots are created. The respective python files are *flagging.py* and *plotting.py*. There is one additional class which is called *ExtractData* and stored in the *extract_data.py* file. It containes functions to collect raw data and processed data files. The software analyses one operation at a time.

The software tool itself is executed by using the *main.py* file. In the *settings_<pine-id>_<campaign>.toml* file, data paths and metadata have to be specified.

In the following paragraphs, the five processing parts are described in detail.

**Lev0Data:**

In this class the raw OPC data are mapped to the respective particle diameters given by the calibration tables and saved as



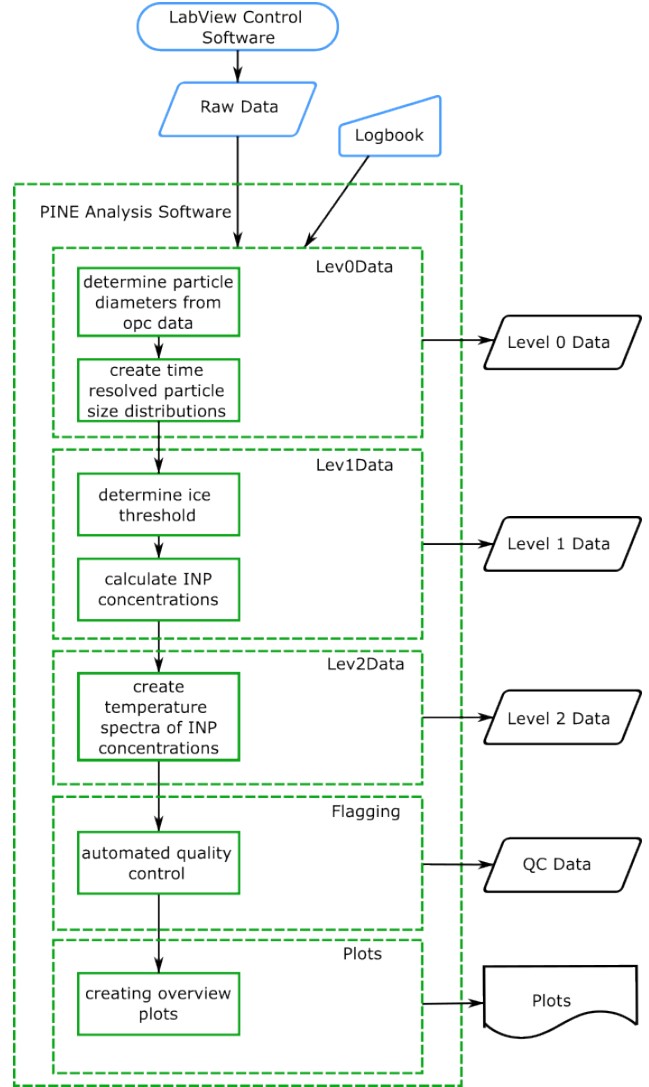

**Figure 2.** Dataflow of Software

Level 0 data. It is important to note that these diameters do not necessarily represent the real diameters of the particles. As

described in Sec. 2.1, the PM voltage of the OPC is adjusted to see cloud droplets and ice crystals, therefore the representation

of the calibration table is not always accurate. As the measured variable of the PINE instrument is the INP concentration, this

does not change anything in the results. The particle diameters are then distributed into time and size bins to allow for easier

handling. The default value for the time bins is 3 seconds, if another time resolution is required this can be changed in the

*settings_analysis_flagging.toml* file. The size bins follow a logarithmically equidistant distribution between 0.1 and 1000 $\mu$m

divided into 100 bins.

**Lev1Data:**





The Level-1-class determines the diameter, above which detected particles are considered to be ice crystals (denoted as ice threshold, see Sect. 4.2). Afterwards, the particle concentration $n_\mathrm{p}$ is calculated as described in Möhler et al. (2021) via

$$n_\mathrm{p} = \frac{c_\mathrm{p}}{F},\qquad(2)$$

with the particle count rate $c_\mathrm{p}$ and the mass flow rate $F$ through the OPC. The INP concentration can be calculated with the same equation (Eq. 2) but using only the particles above the ice threshold. This is done for each time bin and for the whole expansion.

**Lev2Data:**

The Level-2-class creates freezing spectra of the INP concentration, i.e. the INP concentration as a function of their freezing temperature, with a resolution defined as del_temp for each expansion. The default temperature bin width is 0.5 K.

**Flagging:**

This part of the code performs the automated quality control of the data. It creates log-files with errors and warnings for every operation. The applied quality checks are listed in Sec. 4.3.

**Plots:**

The plotting routine is separable from the rest of the code and is independent from the other classes. Three functions for plotting are integrated.

1. Time series: Overview plot for one operation including temperatures, pressure, particle size distribution and INP concentration. Data that are associated with an *error* or *warning* flag are marked in the time series plot.

2. Diagnostics: Creates comparison plots with theoretical dry adiabatic temperatures and saturation ratio for one operation.

3. Histogram: For specified runs, a histogram of the size distribution including the ice threshold is plotted.

**Optional parameters:**

The *main.py* file has two optional parameters. The first one is the "manual_ice_threshold". If it is set to True, the ice threshold will not be calculated, but taken from a file called *<pine-id>_<campaign>_op_id_<operation-id>_ice_threshold_manually.txt*. An example file is shown in the *files*-folder of the software, which also contains the conversion table (*bins_ice_threshold.txt*) from diameter to bin number for the ice threshold. The manual ice threshold files need to be stored in the */L1_Data/exportdata/ice_threshold/*-folder. The second parameter is "cirrus_mode". If this is set to True, the software will not calculate the ice threshold, but the total particle concentration during expansion subtracted by the particle concentration during flush is used as ice concentration. However, measurements at cirrus conditions are not fully tested and applicable for the standard PINE instrument. This feature is mainly implemented for the PINEair prototype (Bogert, 2024).

The output of the analysis software is either stored inside the same folder as the raw data or, if specified in the *settings_<pine-id>_<campaign>.toml* file, at a user-defined path. Table A1 in the appendix summarises the content of the files. All output files start with the prefix <pine-name>_<campaign>_op_id_<op-id>_.





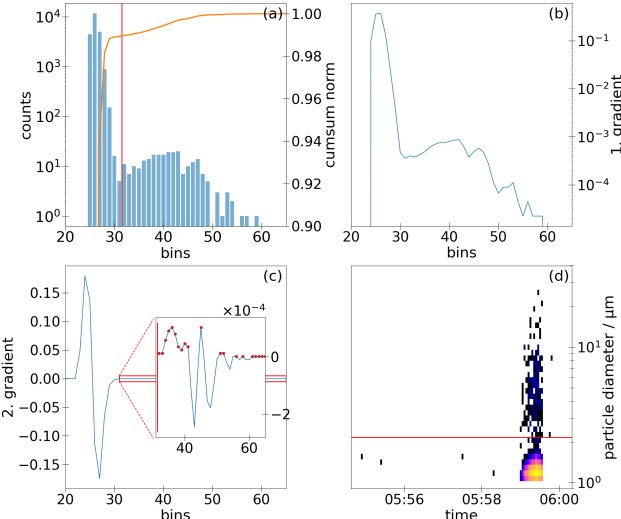

**Figure 3.** Overview plots of the ice threshold finder algorithm. Panel (a) shows the single particle size distribution of one run (in counts on left y-axis) and, in orange, the accumulated and normalised sum of all detected particles (cumsum). Panels (b) and (c) show the first and second gradient of the accumulated sum, respectively. The zoomed-in window in panel (c) shows the part of the second gradient when it becomes positive after the minimum. The red dots mark the values $\geq 0$. Panel (d) shows the time series of the size distributions. The red lines in (a), (c) and (d) indicate the ice threshold diameter. The graph was created using the data of the CORONA campaign, operation 61, run 126.

## 4.2 Ice Threshold Finder

### 4.2.1 Approach

The ice threshold diameter (in the following referred to as the ice threshold) is defined as the particle diameter that distinguishes between ice crystals versus cloud droplets, and aerosol particles, whereby the former are optically larger than the latter. The optical discrimination of particles is based on the size-dependent scattering behaviour of particles, as measured by the OPC.
It is of paramount importance to determine the ice threshold accurately in order to calculate the INP number concentration from the aforementioned measurements. Two principal methods may be employed to determine this threshold diameter. A) From the time series, or B) from the discrete and typically bimodal particle size distribution. On the timescale of PINE expansions, cloud droplets grow by diffusion. The maximum size of cloud droplets, and thus the ice threshold, is impacted by the experimental conditions regarding temperature and supersaturation, as well as the concentration of aerosol particles that serve as cloud condensation nuclei. Due to this, it is preferable to determine this threshold on a run-by-run basis rather than on an operation-wide scale. In addition to the potential for bias at the individual level, the sheer number of expansions renders a manual threshold determination impractical. To address this challenge, a tool has been developed for the PIA software that automatically, objectively, and reproducibly identifies the ice threshold. The particle number size distribution (method B) was





selected as the basis for determining the ice threshold, which corresponds to the minimum between the two modes. Three options for finding this minimum will be discussed in the following sections.

1. Fitting:

The objective is to fit a discrete distribution in order to obtain a continuous probability distribution function (PDF). The automatic fitting of such a function is not a straightforward process. It typically requires either machine learning algorithms or manual adjustments, and it introduces additional uncertainties into the subsequent step of calculating the minimum by differentiating the PDF. This is particularly at odds with the concept of an automated software solution.

2. Discrete calculus:

In the context of discrete distributions, the analogue of the derivative function is the forward differences method. This involves calculating the difference $\Delta f(n)$ between the next and the current bin $n$, as follows:

$$\Delta f(n) = f(n+1) - f(n). \tag{3}$$

A change in sign, from negative to positive, indicates the occurrence of a local minimum. However, this is also the
method's most significant drawback. This method will yield multiple local minima, but no single global minimum. This is also due to two factors:

(a) The OPC size detection range is much larger than the typical size range of aerosols and cloud particles.

(b) The typically few ice particles are spread over a wide size range, resulting in the creation of empty bins.

One potential solution would be to select a smaller size range for the ice threshold finder, where the ice threshold diameter
is typically identified. However, the diameter is contingent upon the experimental conditions (temperature, aerosol type, etc.), rendering the method non-scalable. Moreover, the question of how to distinguish between artificially empty bins in the ice particle mode and the minimum remains unresolved.

3. Cumulative distribution function (CDF):

For each probability distribution, whether discrete or continuous, a continuous monotonic increasing function can be
determined, known as the CDF. In the case of a discrete distribution, the CDF is the bin-wise sum of the distribution, with each bin normalised to a sum of one. The crucial advantage of this approach is its monotone increasing character, which effectively addresses the issue of multiple minima present in option 2. In the case of a bimodal log-normal distribution, the CDF would exhibit a single distinct inflection point, which would directly define the threshold.

Consequently, the automated ice threshold diameter finder tool was developed on the basis of option 3.

**4.2.2  Algorithm**

Based on the aforementioned discussion, the ice threshold finder tool employs the following algorithm:





1. The CDF is calculated by accumulating all size bins over all time bins during flush and expansion mode and normalising it (Fig. 3a orange line), based on the particle number size distribution created in ExtractData for the selected run.

2. From this accumulated and normalised distribution, the first and second gradients (based on the central differences method) are calculated, providing an equivalent to the first and second derivatives of a continuous function (Fig. 3b and 3c).

3. In theoretical terms, the inflection point is defined as the point at which the curvature undergoes a change in sign. Consequently, the second gradient curve crosses the zero line. The algorithm is designed to identify the bins (i.e., the bins above the first minimum in the second gradient CDF) where the second gradient CDF becomes greater than or equal to zero (indicated by the red dots in Fig. 3c).

4. The ice threshold diameter bin is set to the first bin for which the second gradient CDF becomes greater than or equal to zero (Fig. 3d), except in the following cases:

   – No distribution was found.

   – No bins for which the second gradient CDF becomes greater than or equal to zero were identified.

   – The bin of the maximum of the second gradient CDF is larger than the very first bin for which the second gradient CDF becomes greater than or equal to zero (e.g., due to a tri-modal distribution).

   In such cases, the ice threshold bin is set to its default value of 0 (ice threshold = 0.1 μm).

It should be noted that this algorithm is only applicable to mixed-phase clouds. However, if the liquid cloud is not visible in the detection area (e.g., due to an incorrect PM voltage at the OPC), the threshold value cannot be determined. In cases of pure ice clouds (e.g., homogeneous freezing), the algorithm may identify a valid but erroneous ice threshold.

### 4.2.3 Accuracy of Ice Threshold Finder

To estimate the uncertainties introduced by the ice threshold finder, we compared the INP concentrations calculated from manually setting the ice threshold and from the automatically determined ice threshold. The manually set ice threshold is represented by one value for one operation if the temperature is constant. For temperature scans the ice threshold varies slightly within one operation due to the change in temperature. We therefore randomly selected operations conducted at constant temperature and temperature scans.

For comparison, we also included some operations from a laboratory campaign that was conducted under controlled conditions regarding the sampled aerosol population with the model PINE model 05-02 (PINE-05-02) and also operations from a field campaign called CountIce that was conducted by the University of Leeds with their PINE model 04-03 (PINE-04-03), which is from the same production series as the PINE-04-02. Measurements of a laboratory campaign differ from those of a field campaign in the sense that the aerosol type and concentration are controlled in the laboratory. Therefore, much more particles and INPs can be involved. This can be reflected in the bimodal particle size distribution and must be considered when verifying





the ice threshold finder.

The CountIce campaign took place from August 2021 to February 2022 and from March 2023 to August 2023 at the University of Leeds campus (53°48'18.1"N 1°33'10.5"W,  66 m above sea level). These measurements are included for comparison to another PINE model and a different operator setting the manual ice threshold. The comparison was performed for 12 operations from CORONA, 4 operations of the laboratory campaign and 9 operations from CountIce, resulting in a total of more than 4000 analysed runs. For CountIce data above 273 K was neglected.

Figure 4 shows an example of a CORONA operation for which the comparison was carried out. The operation was conducted with a constant wall temperature of about 257 K. It can be seen that the INP concentrations are generally underestimated when using the ice threshold finder. However, there are also some runs for which the INP concentration is overestimated when using the manual threshold (runs at about 15:00).

To quantitatively investigate the uncertainty of the INP concentration, the ratio between the INP concentration calculated from the automatically determined ice threshold and the INP concentration calculated from the manual ice threshold was calculated. This ratio is close to 1 for most of the runs ($\approx 60\%$ have a ratio between 0.98 and 1.02). About 25 % of the runs have a ratio above 1.02, and 15 % fall below a ratio of 0.98.

Figure 5 shows that there is a strong dependency on the total INP concentration with a higher bias at lower INP concentrations. In such cases, when there are only a few ice crystals, the margin of error is larger, as even a slight shift in the ice threshold could result in these low numbers not being counted. It is noticeable that those concentrations derived from the ice threshold finder that are higher than the ones derived from the manual ice threshold are mainly from the CountIce campaign. This difference highlights the need of an objective, operator-independent tool, as the manual ice threshold was set by different responsible persons for each campaign.

## 4.3   Quality Control

All raw and processed data undergo an automated quality control (QC). For the quality control the Python tool System for automated Quality Control (SaQC; Schäfer et al., 2024) is used. SaQC is a tool for automated quality control of time series data that was developed within the Helmholtz Earth and Environment community. The tool has predefined flagging schemes, but also allows users to design a scheme tailored to their own needs.

In case of the PIA software, a scheme returning 3-digit codes for the analysed data was created. Within the software, these codes are mapped to a message containing the run and information about the failed test. The messages including the run number are saved in a log-file.

The flags are reported as failure levels *info*, *warning*, and *error*. The flags, messages, and the respective failure levels are defined in the *settings_analysis_flagging.toml* file. If any data points within a run are flagged with an *error* the run should be excluded from further analysis. If the data are flagged with *info*, a threshold is set in the settings file. This threshold is indicated in the following test descriptions inside brackets behind the flag. If the number of flagged data points within one run exceed the threshold, the data points are flagged as *warning*.

Data with *warning* and *info* flags should be inspected manually, as it may indicate an instrumental malfunction. Therefore, a



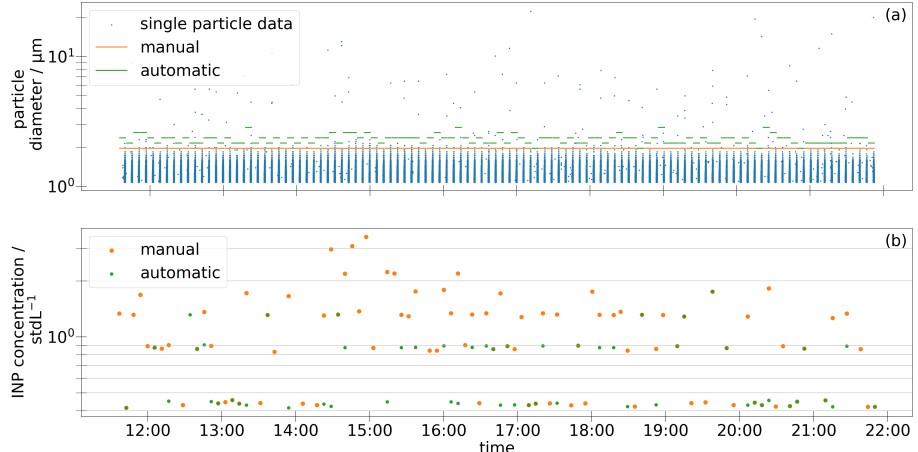

**Figure 4.** Comparison between the manually set ice threshold and the ice threshold determined by the ice threshold finder for operation 140 of the CORONA campaign. The operation was conducted at a constant temperature of about 257 K. Panel (a) shows the single particle data measured by the OPC, the horizontal lines indicate the diameter which represents the ice threshold. Panel (b) shows the INP concentration per run.

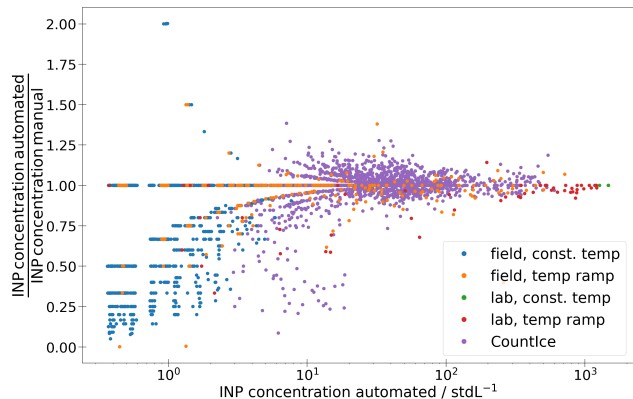

**Figure 5.** Ratio between the INP concentration calculated from the ice threshold determined by the ice threshold finder and the INP concentration calculated from the manually set ice threshold plotted against the total INP concentration derived from the ice threshold finder. The colours show whether the data are taken from a lab or a field study and if the PINE was measuring at constant temperature or performing a temperature scan. Runs with an INP concentration of zero for both manually and automatically determined ice thresholds are displayed with a ratio of one.

manual data reviewing and flagging tool is currently in development. This will help the PINE users to make their data fully quality controlled.

The following two sections describe the tests performed during processing. The tests are split into tests for single data points and tests for a whole run. The latter category describes tests on parameters that influence the result of the whole run: e.g.,





if the ice threshold is not set correctly, the whole run is flagged accordingly. In contrast, an outlier in one of the instrument sensors does not necessarily invalidate the entire run. The detailed information of the test results can be found under */Quality_Control/quality_flags/* in the *flags_instrument.txt* file for the tests per data point and in the *flags_metadata.txt* file for the run-wise tests. Table A2 summarises the QC tests including the internally used flag code, the error type and the short description used in the file headers.

Unless stated otherwise, the test conditions are based on empirical observations.

### 4.3.1 Flags on single data points

These tests are performed on single data points. The zero test on instrument data is performed on the raw data with a 1 second frequency. All other tests are performed on the 3 second averaged data.

**Zero test on instrument data**

The raw data from the instrument file are checked for zero values. If all parameters (temperature, pressure, flow, dew point) have zero values simultaneously, a connection error has occurred. If this is the case, the zero values are replaced with NaNs and handled as missing values for the further analysis.

**Global range test on instrument data**

The instrument data are checked to lie within a valid range for ice formation or according to instrument specifications:

- Ti, Tw: $-60$ to $0\,°C$
- Pch: $495$ to $1055\,mbar$
- Fm during flush and expansion, Fe during expansion: $0.9$ to $5.1\,L/min$
- Fm during refill, Fe during flush and refill: $0$ to $0.02\,L/min$
- DP: $-60$ to $8\,°C$

Flags:

- DP (10), Fe (3), Fm (3): *info*
- Ti1, Ti2, Ti3, Tw1, Tw2, Tw3: *warning*
- Pch, Ti4, Ti5: *error*

**Gaps in instrument data**

If data points are missing due to the zero test or other reasons, it is flagged as *error*.





**Flatline test on instrument data**

This test looks for sensor data, which don't change their value over a certain amount of data point. This could be a result of incomplete data retrieval. The number of consecutive data points with the same value is 5 for the gas temperature, since the temperature decreases during the expansion, and 10 for the pressure and the dew point temperature. The test is performed for:

- Pch

- DP

- Ti during expansion

Flags:

- Ti1 to Ti4, Pch, DP: *warning*

- Ti5: *error*

An example of flatline behaviour in temperature data is shown in Fig. A2.

**Outlier test on flow**

The flow is not constant over all run modes, therefore this test first filters the data for the relevant run modes (for Fe expansion, for Fm flush and expansion) and performs a Z-test on the flow rate. If the first or last data points of a run mode are flagged, they are ignored, as the flow typically stabilises after a few seconds. Other outliers are flagged as *warning*.

**Global range test on particle concentration**

This checks if the particle concentration lies within a plausible range. The range is 0 to $700'000\,\text{stdL}^{-1}$. Data outside this range are flagged as *warning*.

**Particles during refill**

During refill, no flow is actively passing the OPC, so no particles should be detected. The first two time bins of the refill mode are ignored, as it is possible that there are still particles in the line after the pump is turned off. If particles are detected in any other bins, the data are flagged as *warning*. This could be a sign for a contaminated OPC or icing inside the chamber. Figure A3 shows an example run where particles were detected during refill.

### 4.3.2 Flags on runs

These tests are performed once per run. If they fail, the whole run is flagged.

**Test on run times**



This test checks that the run times (start, expansion, refill, end) are valid and strictly increasing. If this test fails, the run is flagged with *error*.

**Missing OPC data**

If there is no data saved to an *opc file*, it is assumed that there was a connection problem with the OPC. In this case the data points are flagged with *error*.

**Variability of ice threshold**

For this test, the mean and standard deviation of all ice thresholds in one operation are calculated. If a single ice threshold

deviates more than two standard deviations from the mean, the run is flagged with a *warning*.

**Large aerosol particles during flush mode**

If, during flush mode, more particles above the ice threshold than a certain concentration threshold are detected, the run is flagged with *warning*. This indicates a high ice background inside the chamber. The threshold for the INP concentration during

flush mode is 50 % of the INP concentration during the expansion.

**Comparison to pre-set values**

For each run the end pressure during the expansion and the mean flow during flush and expansion are set and stored in the run file. The measured values are then compared to the pre-set values. The following ranges are accepted:

– Pch: $\pm 5\,\%$

    – Fm, Fe: $\pm 15\,\%$

If these tests fail, the run is flagged as *warning*.

**Temperatures at the end of expansion**

The temperatures sensors inside the chamber should have the following order at the end of the expansion: Ti1 > Ti2 > Ti3 > Ti4 > Ti5. If this is not the case for the temperature sensors Ti1 to Ti4, the data are flagged with *warning*. If Ti5 is higher than Ti4, it results in an *error*, as Ti5 is used as the reference temperature for the INP concentration. If this is the case it may be due to a malfunction of the temperature sensors or a wrong positioning of the sensors inside the chamber. Figure A4 shows an example of expansions where Ti5 measures a higher value than Ti4.

**Global range test for duration of expansion**

The duration of the expansions must lie within the following ranges:

    – Ti5 below $-35\,°C$: 9 sec to 2 min





– Ti5 above $-35\,°\mathrm{C}$: 20 sec to 2 min

If this test fails the run is flagged with *error*.

**Check for supersaturation**

If supersaturated conditions are not given inside the chamber, ice cannot form. This is tested by comparing Ti5 to DP at the beginning of the expansion. If Ti5 is higher than DP, supersaturation is likely not achieved, and the run is flagged. As the DP

sensor is positioned in front of the chamber, it is still possible to have supersaturation, even if Ti5 is higher than DP, namely when an ice layer formed on the chamber wall after a longer time of operation. Thus this check is only flagged as *warning*.

**Icing of chamber inlet**

If the sampled air is highly humid, ice may form on the chamber inlet. This reduces the flow rate into the chamber and, con-

415 sequently, the pressure inside the chamber during flush. The pressure sensor is positioned outside the chamber, in front of the iced inlet, so it does not detect this reduced pressure. Once the valve closes, the pressure equilibrates, which is reflected by a faster-than-expected pressure drop at the start of expansion. To detect such cases, the theoretical expansion duration is calculated from the initial pressure at the beginning of the expansion $P_0$, the pre-set end pressure $P_{\mathrm{end}}$, the pre-set expansion flow $Fe_{\mathrm{set}}$, and the volume of the chamber $V_{\mathrm{ch}}$:

$$\Delta t_{\mathrm{exp}} = \frac{\frac{P_0}{P_{\mathrm{end}}} \cdot V_{\mathrm{ch}} - V_{\mathrm{ch}}}{Fe_{\mathrm{set}} \cdot 60\,\mathrm{s}}. \tag{4}$$

If the actual expansion duration is shorter than the theoretical value, the run is flagged with *error*. This test only works if a certain threshold is exceeded. Runs that are only slightly affected by the iced inlet are not detected. To ensure no faulty runs are missed, all runs following three consecutively flagged runs, as well as the five preceding them, are also flagged. If five or fewer unflagged runs remain in an operation, the entire operation is flagged. Figure A5 shows an example of pressure measurements

during inlet icing.

**Gaps in OPC data**

This tests checks for missing data in the OPC data during expansion. This is first applied to 3 second binned data. If a bin without OPC data is found, the test is repeated with 1 second bins. This test is only performed if the total particle concentration

of the run exceeds $180\,\mathrm{stdL^{-1}}$. If the particle concentration is lower, gaps seen in the OPC data are most likely real gaps. If this test fails, the run is flagged with *error*.



## 4.4  Maintenance of Software

The PIA software is managed in a publicly available GitLab repository[1] containing multiple branches. The main branch is used
for releases and the development is done in the development branch, or in branches dedicated to a specific task. The issues-
system is used to mark needed improvements, bug reports, or new feature requests. The issues can be filed from code developers
or PINE users. The repository has a Continuous Integration and Continuous Delivery (CI/CD) pipeline implemented, which
runs multiple tests for every push event to the repository. These tests include: linting tests, check that all licensing information
is conform with the REUSE (Free Software Foundation Europe, 2025) recommendations, verifying that the software is running
for different PINE and Python versions. For the latter, test data from different PINE and LabVIEW control software versions are
included in the repository to ensure their compatibility with new PIA software or Python versions. Since PIA version 3.1.0, the
CI/CD pipeline includes a workflow for automated releases on Zenodo, based on the FACILE-RS tool (openCARP consortium
et al., 2025). This workflow is triggered by creating a *pre*-tag on the main branch. It updates the metadata, the changelog-file,
and the version number. Afterwards, the real version tag and a release on GitLab is created, followed by pushing the code
and the metadata to Zenodo, where it can be released after manual inspection. The documentation for the PIA software[2] is
hosted on Read the Docs (Read the Docs, Inc & contributors, 2025). The GitLab repository is connected to Read the Docs via
a webhook, and whenever a new tag in the repository is created, the documentation will be updated.

With regard to the above mentioned workflow, the PIA software follows the FAIR4RS (FAIR for Research Software; Barker
et al., 2022) principles. The fulfilment of these principles is summarised with the FAIR software checklist[3].

## 5  Quality Assurance of PINE Measurements

### 5.1  Characterisation of Temperature Sensors

The gas temperature inside the chamber is influenced by the ambient air entering from the top. Therefore, a temperature gradient
exists, depending on the temperature of the incoming air and the wall temperature, with the highest temperatures at the top and
the lowest temperatures at the bottom of the chamber. This general temperature gradient is also observed during expansion:
while the gas temperature decreases at all sensor locations, the lowest temperature is consistently recorded by the sensor at the
lowest position (Ti5). Evaluation of the behaviour of the temperature sensors can help to validate the functionality of the sensors
and identify possible malfunctions. To verify the stability and comparability of this behaviour over time and across different
instruments, the temperature differences between neighbouring sensors were analysed for three different PINE models and 22
measurement campaigns. These differences were inspected for correlations with the pressure change during expansion and
the end temperature of Ti5 during expansion. The run-wise data did not show any significant correlations. Nevertheless, it is
interesting to compare the different PINE models in order to identify any systematic deviations in the reported temperatures.
Figure 6 shows the temperature difference between sensors Ti3 and Ti4, and between sensors Ti4 and Ti5 as a function of

---

[1]https://codebase.helmholtz.cloud/pine/pia_software

[2]https://pia.readthedocs.io/en/stable/

[3]https://fairsoftwarechecklist.net/v0.2?f=31&a=32113&i=22101&r=123





the temperature measured at the end of the expansion. Ti1 and Ti2 usually show high fluctuations, which can be expected as they are the those mostly impacted by the entering ambient air, and are therefore not further analysed. The data is temperature binned per PINE model and the markers show the mean values, while the error bars show the standard deviations. The data of the PINE-05-02 is split into two. because the temperature sensors had to be repositioned by the manufacturer, as Ti5 was positioned to close to the wall and therefore reporting higher values than Ti4 due to the wall temperature. The values from before this repositioning are marked in red, the values marked in green are results from afterwards. While the differences between Ti3 and Ti4, and Ti4 and Ti5 agree well for the PINE models 04-01 and 04-02, the results for PINE-05-02 (after repositioning) show a different behaviour. The difference between Ti4 and Ti5 is about 1 K higher than for the other instruments and the difference between Ti3 and Ti4 is slightly lower than for the other instruments. This could indicate that the sensors Ti4 and Ti5 are at different positions in PINE-05-02 compared to the positions in the other PINE models. It is possible that the position of the Ti5 sensor is located in a colder region of the chamber compared to the Ti5 positions in other instruments and that the Ti4 sensor is positioned in a warmer region.

In the future, a better control and documentation of the sensor positioning is required. This would enable the use of temperature differences between sensors as a quality assurance measure. This test would be particularly useful for the initial validation of new instruments.

## 5.2 Background Measurements

Ice can form on the chamber walls during continuous operation of PINE, if the air inside the chamber is too humid. To ensure that this does not influence the INP concentration, regular ice background measurements must be performed. Therefore, ambient air is guided through a HEPA filter to ensure that no external particles enter the chamber, meaning that any detected cloud droplets or ice crystals are only due to particles already inside the chamber. These operations must be marked as background measurements in the electronic logbook.

Analysis of background measurements from existing campaigns shows that it takes approximately 80 to 100 L to completely flush out the aerosol inside the chamber. This includes both the air that is guided through the chamber during flush and the air filled into the chamber during refill.

We recommend performing background measurements once per day using the following settings (or similar, according to the settings during measurement operations) to achieve an air exchange of at least 80 L: 9 runs with 5 min flush time at a mass flow rate of $2\,\text{stdL}\,\text{min}^{-1}$, and a pressure reduction during expansion to approximately 15 % below ambient conditions.

If the ninth run still shows significant particle counts, the chamber is likely iced or otherwise contaminated. Ice-sized particles should not be visible after the sixth run.

## 5.3 Influence from Large Aerosol Particles

Aerosols with diameters larger than the ice threshold are occasionally observed during flush mode. For the CORONA campaign, this occurred in more than 30 % of the runs. If such large aerosols are present during flush, it is possible that they are also present during expansion and contribute falsely to the INP concentration. Figure 7 shows the ratio of the ice-sized aerosol concentration





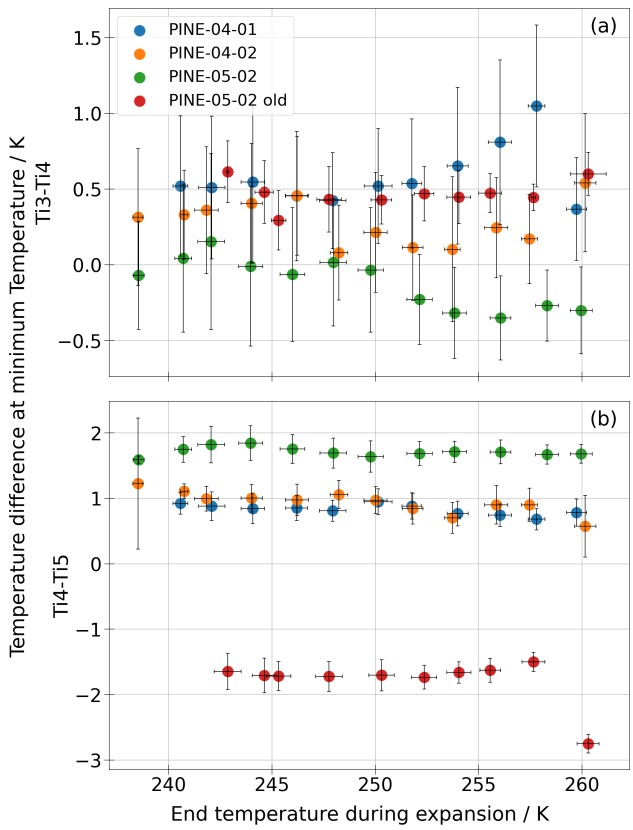

**Figure 6.** Difference between two neighbouring temperature sensors as a function of the temperature measured at the end of the expansion. The values are temperature binned and the error bars show the standard deviations of the x- and y- values. Different colours indicate different instruments.

during flush and the INP concentration during expansion as a function of the INP concentration during expansion. For INP concentrations above $10\,\mathrm{std L^{-1}}$, the influence is mostly negligible, but it becomes more relevant at lower concentrations. This influence may in fact be smaller, as the size of aspherical particles appears larger to the OPC. If water condenses on these particles, they become spherical droplets and appear optically smaller. More systematic measurements need to be performed to understand this potential influence from large aerosol particles. For now, large aerosol particles should be accounted for by including the ice-sized aerosol concentration during flush in the measurement uncertainties, if they exceed the statistical uncertainty. These values are also stored in the *ice.txt*-files generated by the PIA software.





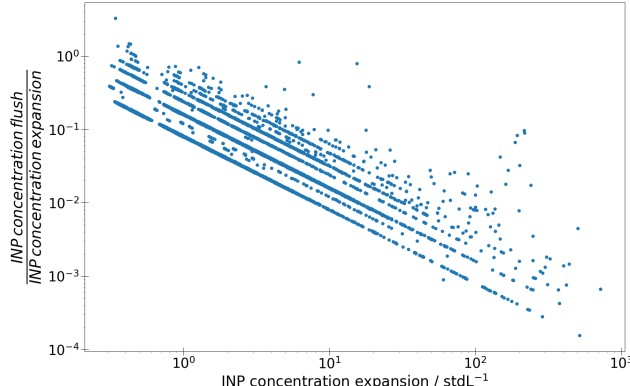

**Figure 7.** Ratio of ice-sized aerosol concentration during flush and INP concentration during expansion as a function of the INP concentration during expansion.

## 6 Community Aspects

### 6.1 Aerosol, Clouds, and Trace Gases Research Infrastructure (ACTRIS)

The ACTRIS (Laj et al., 2024) Topical Centre CIS is one of the six Topical Centres, supporting the observational platforms in producing high-quality data and information on the abundance, variability, and trends of short-lived atmospheric constituents and related processes. Among the variables for CIS are the concentration and freezing spectrum of INPs. CIS provides measurement guidelines for operating the online instruments PINE and the automated CFDC Horizontal Ice Nucleation Chamber (HINC-Auto; Brunner and Kanji, 2021), as well as the offline INP measurement method INSEKT. The PINE instrument serves for laboratory-based validation and for field-based intercomparisons, aiming to achieve accurate and quality-controlled long-term INP data at ACTRIS observational platforms, atmospheric simulation chambers, and mobile platforms.

### 6.2 PINE Community Portal

A growing worldwide community of PINE users will also be connected to the future INP monitoring network and CIS activities in Europe. To strengthen the INP community and enhance collaborations, we developed the PINE community portal[4]. The portal is publicly accessible and maintained by KIT. Any user is able to submit additional datasets to the portal. Existing data can be viewed online and up to three measurement sites can be compared across different nucleation temperatures.

### 6.3 Application for other PINE Versions

Since PIA software version 3.0.0, the compatibility with the further developed prototype PINEair and the laboratory-based instrument AIDAm (Vogel et al., 2022) is given. Thanks to its well-defined data structure, comprehensive data acquisition software and modular architecture, the PIA software can easily be adapted for future PINE versions.

---

[4]https://pcp.imk.kit.edu/pine-dashboard



# 7 Conclusions

A standardised workflow for handling data from the PINE instrument was established. Therefore, the Python software package PIA (PINE INP Analysis) was developed and is presented here. The software analyses the raw data acquired by an in-house developed LabVIEW control software and converts it into time series of INP concentrations and freezing spectra. The required ice threshold for distinguishing between aerosols and droplets versus ice crystals is automatically determined by the ice threshold finder described in Sect. 4.2. This allows for a standardised data set that is independent of the data analyst. All data undergoes several automated quality control tests, which are described in detail in Sect. 4.3. These tests are continuously reviewed by the PINE users and will be further developed as needed. The same applies for the ice threshold finder. As the number of data sets increases, statistical robustness will improve, allowing the current algorithm to be verified or further improved. The modular structure of the software allows for easy adaption to future PINE versions.

For further improving the software, the possibility to manually flag the data will be implemented in the future. It is also desirable that the LabVIEW control software automatically generates the metadata currently provided by the manual electronic logbook (e.g., sample aerosol, operation type) and stores it in the raw data to ensure compatibility with the PIA software and further increase the automatisation of the data workflow.

Section 5 provides recommendations on how to treat the data and conduct measurements to ensure reliable and standardised results. The offset between temperature sensors should be checked as part of the initial validation and possibly after longer transport of the instrument, as an indicator of sensor positioning inside the chamber. Background measurements should be performed regularly (approximately once per day) to check for icing and contamination inside the chamber. The presence of large aerosol particles during flush mode should be considered when calculating measurement uncertainties, if they exceed the statistical uncertainty given by Eq. 1.

*Code and data availability.* The PIA software is available on Zenodo (https://doi.org/10.5281/zenodo.15592883).

The CORONA data set is available on RADAR4KIT (https://dx.doi.org/10.35097/sqmdyj7ckbccq9zy).

The CORONA_new data set is available on RADAR4KIT (https://doi.org/10.35097/c78mxhyjyd269pr9).

The CountIce data set is available on zenodo (https://doi.org/10.5281/zenodo.17451019). For this work the data sets 211015, 211110, 211122, 211214, 220110, 230316, 230414, 230524, and 230609 have been used.

# Appendix A

# A1





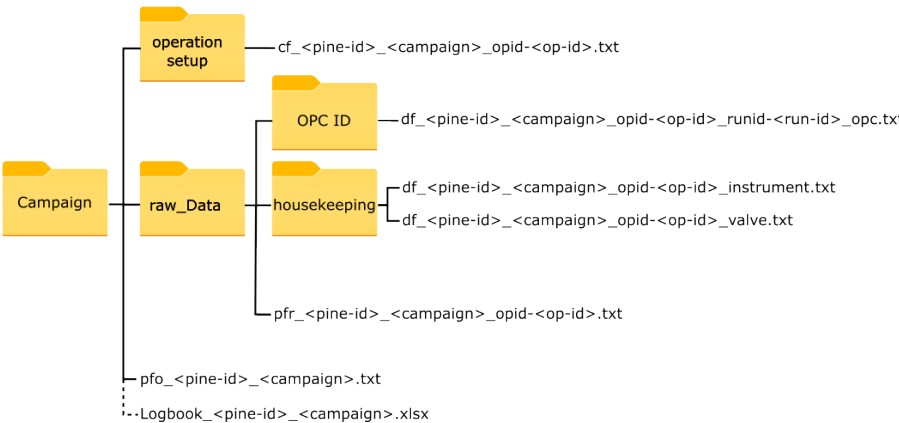

**Figure A1.** Raw data structure as produced by the LabVIEW control software. The dashed line indicates the location of the manual electronic logbook, which is not created by the LabVIEW control software.

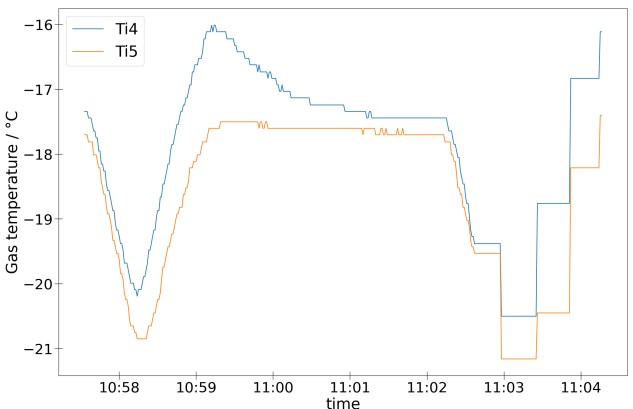

**Figure A2.** Example of flatlines in the instrument data. The plot shows operation 76, run 16 from the CORONA_new campaign.

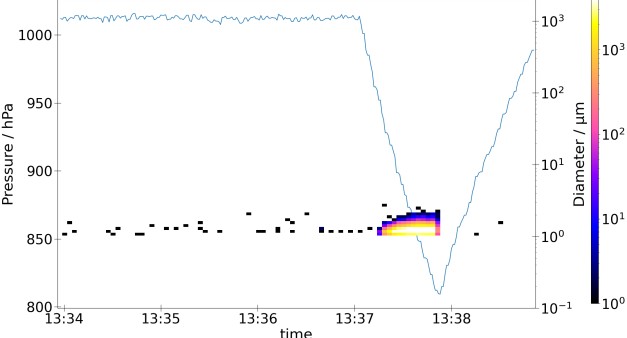

**Figure A3.** Example of particles detected during refill mode. The plot shows operation 28, run 61 from the CORONA_new campaign.



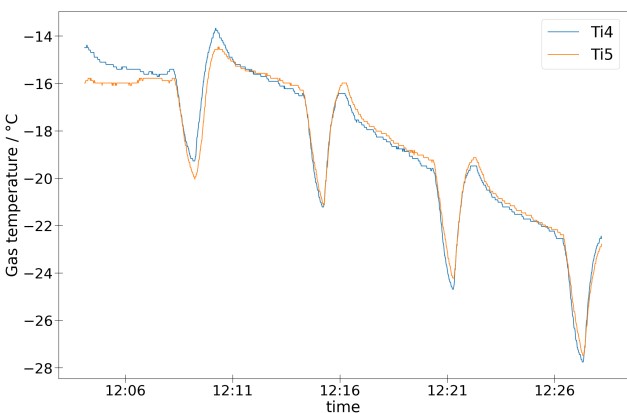

**Figure A4.** Example of Ti4 measuring a lower temperature than Ti5. The plot shows operation 205, runs 4 to 7 from the CORONA campaign.

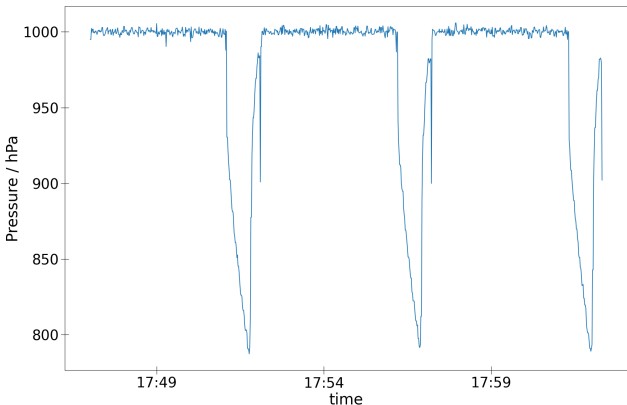

**Figure A5.** Example of detected inlet icing. At the start of the expansion a steep drop can be seen in the pressure. The plot also shows the characteristic pressure drop at the end of the refill. The plot shows operation 198, runs 10 to 12 from the CORONA campaign.





**Table A1.** Output files

| file ending | location | content | metadata |
|---|---|---|---|
| opc_spd.txt | /L0_Data/exportdata/ exportdata_opc_spd/ OP<op-id>/ | The single particle data from the raw OPC data with an additional column containing the diameter representing the bin number. | software version, description of variables |
| cn.txt | /L1_Data/exportdata/ exportdata_cn/ OP<op-id>/ | 3 second binned data: time relative to start of run, particle concentration, particle count, INP concentration, INP count, run mode: flush (1), expansion (2), refill (3), mean flow rate, mean chamber pressure, mean Ti1, mean Ti2, mean Ti3, mean Ti4, mean Ti5, particle counts per bin size | software version, station, latitude, longitude, altitude, PI, institution, aerosol, operation type, OPC number, start time, flush duration, size of time bins, ice threshold, description of variables |
| ice.txt | /L1_Data/exportdata/ exportdata_ice/ | run-wise data: run ID, time at the middle of the flush, minimal temperature (Ti5) reached in expansion, pressure at the end of expansion, mean flow rate during expansion, INP concentration during expansion, INP concentration during flush, particle concentration during expansion, particle concentration during flush | software version, station, latitude, longitude, altitude, PI, institution, aerosol, operation type, OPC number, description of variables |
| ice_threshold.txt | /L1_Data/exportdata/ ice_threshold/ | run-wise data: run ID, bin number, ice threshold diameter | software version |
| temp.txt | /L2_Data/Temp_Spec/ | temperature-binned data: start temperature, end temperature, run ID, time at the middle of the flush, mean time relative to start of expansion, mean INP concentration, lowest chamber pressure | software version, station, latitude, longitude, altitude, PI, institution, aerosol, operation type, OPC number, description of variables |
| temp_mean.txt | | operation mean of temperature-binned data: start temperature, end temperature, mean INP concentration in temperature bin over all runs, standard deviation of INP concentration | software version, station, latitude, longitude, altitude, PI, institution, aerosol, operation type, OPC number |
| flags_instrument.txt | /Quality_Control/ quality_flags/ | flags for data points, for internal use of the software | software version |
| flags_metadata.txt | /Quality_Control/ | flags for tests per run, for internal use of the software | software version |
| flags.log | | Log file with warnings and errors for all runs | |



**Table A2.** Overview of qc tests, including the software internal flags, the error type and the short description used in the headers of the *flags_instrument.txt* and *flags_metadata.txt* files.

| test | flag | type | short |
|---|---|---|---|
| Fe outlier | 459 | info/warning | Fe_outlier |
| Fm outlier | 460 | info/warning | Fm_outlier |
| particle concentration range | 461 | warning | concentration_range |
| variability ice threshold | 463 | warning | ice_threshold_2_sigma |
| supersaturation | 549 | warning | supersaturation |
| difference between Ti1 and Ti2 | 635 | warning | Ti1/Ti2_diff |
| difference between Ti2 and Ti3 | 636 | warning | Ti2/Ti3_diff |
| difference between Ti3 and Ti4 | 637 | warning | Ti3/Ti4_diff |
| difference between Ti4 and Ti5 | 638 | error | Ti4/Ti5_diff |
| comparison to pre-set Fe, Fm | 664, 665 | warning | Fe, Fm |
| run times | 670 | error | increasing_time |
| icing of inlet | 677 | error | dur_expansion |
| Pch range | 684 | error | Pch_range |
| DP range | 685 | info/warning | DP_range |
| Fe range | 686 & 688 | info/warning | Fe_range & Fe_range_zero |
| Fm range | 687 | info/warning | Fm_range |
| Ti1 - Ti3 range | 689 - 691 | warning | Ti1_range - Ti3_range |
| Ti4, Ti5 range | 692, 693 | error | Ti4_range, Ti5_range |
| Tw1 - Tw3 range | 694 - 696 | warning | Tw1_range - Tw3_range |
| DP flatline | 698 | warning | DP_flatline |
| Pch flatline | 699 | warning | Pch_flatline |
| Ti1 - Ti4 flatline | 700 - 703 | warning | Ti1_flatline - Ti4_flatline |
| Ti5 flatline | 704 | error | Ti5_flatline |
| duration of expansion | 708 | error | dur_expansion |
| comparison to pre-set Pch | 709 | warning | pch_end |
| particles during refill | 710 | warning | opc_refill |
| large aerosols during flush | 711 | warning | inp_flush |
| missing opc data | 997 | error | concentration_missing |
| gaps in opc data | 998 | error | opc_data |
| gaps instrument data | 999 | error | — |



*Author contributions.* NB prepared the manuscript with contributions from RF, OM, LL and FV. The software was designed and written by
NB and RF. LL, FV, and MT reviewed the data for the manual ice threshold analysis. AB, LL, FV, PB, and JN contributed to the design of
the quality control tests and the output data. FV, LL, OM, MT, and BM were responsible for collecting and processing the observational data.

*Competing interests.* The authors declare that they have no conflict of interest.

*Acknowledgements.* AI has been used to check the manuscript for grammatical errors and improve its formulation.
CountIce was supported by the European Research Council Proof of Concept grant (862565). The development of the PINE instrument was
supported by the Karlsruhe Institute of Technology through the technology transfer project "A Portable expansion chamber for Ice Nucleating
particle mEasurements - PINE (project number N059)" and by the University of Leeds International Strategy Fund. The development of the
automated quality control was supported by the Digital Earth project of the Helmholtz Association.



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
