# Peer review of "Automated Analysis and Quality Assurance of Ice-Nucleating Particle Data: The PINE INP Analysis Software PIA"

_EGUsphere, 2025_

## Referee Comment (RC1)

This manuscript presents the PINE INP Analysis (PIA) software version 3.0.0. The development of standardized, automated, and open tools for processing complex instrumental data is essential for ensuring data quality, reproducibility, and inter-comparability across studies. The manuscript is well-structured and clearly written, providing a comprehensive overview of the PIA software's architecture, core functionalities, and its role in supporting high-quality INP measurements. I recommend publication after minor revisions.

1. Fig. 3: The particle size distribution of a single run. What is the total duration of one run?

2. Section 5.3: Influence of Large Aerosol Particles. If large aerosol particles are present, will a two-mode size distribution be observed? Assuming a field campaign with a strong dust plume (coarse mode particle number concentration exceeding 40 # $cm^{-3}$), will the two-mode ice threshold finder function effectively?

3. Should quality control (Sect. 4.3) be performed before using the ice threshold finder (Sect. 4.2)?